# PERIL: PROBABILISTIC EMBEDDINGS FOR HYBRID META-REINFORCEMENT AND IMITATION LEARNING

## ABSTRACT

Imitation learning is a natural way for a human to describe a task to an agent, and it can be combined with reinforcement learning to enable the agent to solve that task through exploration. However, traditional methods which combine imitation learning and reinforcement learning require a very large amount of interaction data to learn each new task, even when bootstrapping from a demonstration. One solution to this is to use meta reinforcement learning (meta-RL) to enable an agent to quickly adapt to new tasks at test time. In this work, we introduce a new method to combine imitation learning with meta reinforcement learning, Probabilistic Embeddings for hybrid meta-Reinforcement and Imitation Learning (PERIL). Dual inference strategies allow PERIL to precondition exploration policies on demonstrations, which greatly improves adaptation rates in unseen tasks. In contrast to pure imitation learning, our approach is capable of exploring beyond the demonstration, making it robust to task alterations and uncertainties. By exploiting the flexibility of meta-RL, we show how PERIL is capable of interpolating from within previously learnt dynamics to adapt to unseen tasks, as well as unseen task families, within a set of meta-RL benchmarks under sparse rewards.

## 1 INTRODUCTION

Reinforcement Learning (RL) and Imitation Learning (IL) are two popular approaches for teaching an agent, such as a robot, a new task. However, in their standard form, both require a very large amount of data to learn: exploration in the case of RL, and demonstrations in the case of IL. In recent years, meta-RL and meta-IL have emerged as promising solutions to this, by leveraging a meta-training dataset of tasks to learn representations which can quickly adapt to this new data. However, both these methods have their own limitations. Meta-RL typically requires hand-crafted, shaped reward functions to describe each new task, which is tedious and not practical for non-experts. A more natural way to describe a task is to provide demonstrations, as with meta-IL. But after adaptation, these methods cannot continue to improve the policy in the way that RL methods can, and are restricted by the similarity between the new task and the meta-training dataset. A third limitation, which both methods can suffer from, is that defining a low-dimensional representation of the environment for efficient learning (as opposed to learning directly from high-dimensional images), requires hand-crafting this representation. As with rewards, this is not practical for non-experts, but more importantly, it does not allow generalisation across different task families, since each task family would require its own unique representation and dimensionality.

In this work, we propose a new method, PERIL, which addresses all three of these limitations, in a hybrid framework that combines the merits of both RL and IL. Our method allows for tasks to be defined using demonstrations only as with IL, but upon adaptation to the demonstrations, it also allows for continual policy improvement through further exploration of the task. Furthermore, we define the state representation using only the agent's internal sensors, such as position encoders of a robot arm, and through interaction we implicitly recover the state of external environment, such as the poses of objects which the robot is interacting with. Overall, this framework allows for learning of new tasks without requiring any expert knowledge in the human teacher.

PERIL operates by implicitly representing a task by a latent vector, which is predicted during learning of a new task, through two means. First, during meta-training we encourage high mutual information between the demonstration data and the latent space, which during testing forms a prior for

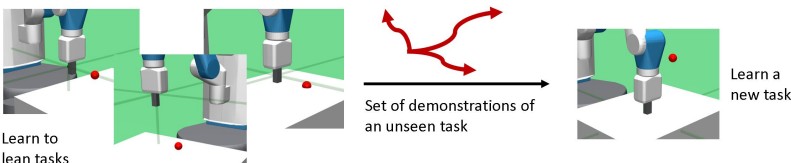

Figure 1: Overview of our proposed method. We obtain a set of demonstrations of an unseen task and we adapt to it through efficient demonstration-conditioned exploration.

the latent space and represents the agent's task belief from demonstrations alone. Second, we allow further exploration to continually update the latent space, by conditioning on the robot's states and actions during exploration of this new task. We model the latent space via an encoder, from which posterior sampling can be done to encode the agent's current task belief. In essence, the encoder aims to learn an embedding which can simultaneously (i) infer the task intent, and (ii) output a policy which can solve the inferred task. During meta-training, PERIL is optimised end-to-end, by simultaneously learning both a policy and the embedding function upon which the policy is conditioned.

In our experiments we find PERIL achieves exceptional adaptation rates and is capable of exploiting demonstrations to efficiently explore unseen tasks. Through structured exploration, PERIL outperforms other Meta-RL and Meta-IL baselines and is capable of zero-shot learning. We show how our method is capable of multi-family meta-learning as well as out-of-family meta-learning by clustering distinct meta-trained latent space representations. As an extension, we also show how to use privileged information during training to create an auxiliary loss for training the embedding function, which helps to form a stronger relationship between the latent space and the true underlying state which defines the task. Supplementary videos are available at our anonymous webpage https://sites.google.com/view/peril-iclr-2021

## 2 RELATED WORK

Meta-RL was conceptualised as an RNN-based task. Developed by Wang (2016) and Duan et al., the authors use RNNs to feed a history of transitions into the model such that the policy can internalise the dynamics. On another line, Finn et al. (2017) developed a learning-to-learn strategy, model agnostic meta learning (MAML), which meta-learns an initialisation that adapts the parameters of the policy network and fine tunes it during meta-testing. Although promising results in simple goal-finding tasks, MAML-based methods fail to produce efficient stochastic exploration policies and adapt to complex tasks (Gupta et al., 2018). Meta-learning robust exploration strategies is key in order to improve sample efficiency and allow fast adaptation at test time. In light of this, context-based RL was developed with the aim of reducing the uncertainty of newly explored tasks. These map transitions $\tau$ collected from an unseen task into a latent space $z$ via an encoder $q_\phi(z|\tau)$, such that the conditioned policy $\pi_\theta(\tau|z)$ can efficiently solve said task (Rakelly et al.; Wang & Zhou, 2020). An underlining benefit of decoupling task encoding from the policy is that it disentangles task inference from reward maximisation, whilst gradient-based and RNN-based meta-RL policies do this internally. An important remark regarding training conditions of the discussed meta-RL methods is that they are typically meta-trained using dense reward functions (Rakelly et al.; Wang, 2016; Wang & Zhou, 2020). These dense reward functions provide information-rich contexts of the unseen task. Considering that the ultimate goal is to allow agents to solve new tasks in the real world, adaptation during test time must be robust to sparse reward feedback (Schoettler et al., 2020).

In the context of RL, incorporating demonstrations has proven successful in aiding exploration strategies, stabilising learning and increasing sample efficiency (Vecerik et al., 2017). Learning expressive policies from a set of demonstrations requires a vast amount of expert trajectories (Ross et al.), particularly in high dimensional state-action spaces (Rajeswaran et al.). In contrast, Meta-IL can be implemented in meta-RL by conditioning the agent with expert trajectories. On the another hand, Zhou et al. propose a MAML-based meta-IL approach which averages objective across demonstrations. The latter learns to adapt at test-time by receiving binary rewards. A similar strategy was also developed by and Mendonca et al.. The caveats of these approaches remain that of traditional IL: (i) Imitating expert trajectories hinders the policy from doing better than the demonstrations; (ii) Cloning behaviours reduces flexibility and generalisation capacity.

## 3 Hybrid Meta-Reinforcement and Imitation Learning

### 3.1 Problem Statement

We assume access to a set of tasks $\mathcal{T} \in p(\mathcal{T})$, with each task represented as a partially observable Markov decision process (POMDP). Observations $o$ are incomplete and only include measurements of the agent's internal state. For example, a peg-in-hole task would include the robot's (and therefore the peg's) pose in the observation, but not the hole's pose. The full state $s$ contains both the robot's internal state, and the state of the external environment. Our method involves an agent inferring $z$, a descriptor of a task which provides the information required to solve that task, such as a representation of $s$. Each task is then defined as $\mathcal{T} = \{p(o_0|z), p(o\prime, z\prime|o, a, z), r(o, a|z)\}$, with an unknown initial observation distribution $p(o_0|z)$, transition distribution $p(o\prime, z\prime|o, a, z)$, and reward function $r(o, a|z)$, where $a$ is the action taken by the agent. By leveraging task beliefs, PERIL exploits enriched observational spaces ($s \sim \{o \cap z\}$) with the objective of closing said POMDP into a stable MDP form. With the aim of supporting continuous task inference through interaction, we construct $z$ as a probabilistic embedding conditioned on a set of recently collected transitions, referred to as the context $c$. We define a contexton $c_t^{\mathcal{T}} = (o_t, a_t, r_t, o_{t+1})$ as a transition collected on task $\mathcal{T}$ at time-step $t$ such that the context $c_{0:t}^{\mathcal{T}}$ (what we call $c$) denotes the set of accumulated contextons. During meta-testing, the agent then attempts to find the true posterior $p(z|c)$ over the task belief $z \sim p(z|c)$, by conditioning on $c$.

Demonstrations significantly reduce the search space in exploration whilst providing a natural means of communicating the task. Thus, access to expert trajectories provides information-rich context which can be exploited to pre-condition the policy via a prior over $z$. Subsequently, online adaptation through RL can further disambiguate $z$. In our approach, we leverage dual meta-learning objectives to perform hybrid adaptation to new tasks, with conditioning on both demonstration contexts and exploratory contexts, where we define a demonstration as a trajectory of $T$ observations and actions $d = \{(o_t, a_t)\}_{t=1}^{T}$, and an adaptation episode $\tau_e$ as a trajectory of $N$ steps $\tau_e = \{(o_t, a_t, r_t, z_t)\}_{t=1}^{N}$:

**Primal Inference:** The agent observes a set of $k = 1, ..., K$ demonstrations from an expert set $D_{demo} := \{d_k\}$ to form a context prior $c_{demo} = \{d_k\}_k^K$.

**Exploratory Adaptation:** The agent explores with initialised context $c \leftarrow c_{demo}$ and adapts with sampled trajectories $\tau$ such that at time-step $t$, a hybrid context is formed $c \leftarrow c_{demo} \cap \tau_{0:t}$.

Primal task inference aims to provide long-term inference on a task belief: meta-trained inference from previously solved tasks. Using posterior sampling, exploratory adaptation provides a proxy for short-term memory: as the context is updated with recent transitions, structured exploration can adjust primal inference by reasoning about the uncertainty of $z$, where $c := \{c_{demo} \cap (\tau_e^T, ..., \tau_e^{\infty T})\}$.

### 3.2 Probabilistic Embeddings for Meta-RL

As we do not have access to the posterior $p(z|c)$, we use variational inference methods to produce an approximation $q_\phi(z|c)$. Through generative processes, we sample $z \sim q_\phi(z|c)$ and optimise $\phi$ by maximising a meta-objective conditioned on $z$. Variational Encoders (VEs) are capable of producing latent distributions which optimise an arbitrary task-dependent objective $\mathcal{G}(\mathcal{T}|\cdot)$. We average the expectancy over a set of tasks from $p(\mathcal{T})$ to formulate a meta-objective which rewards fast adaptation to unseen tasks (Eq. 1), where $\beta$ controls the constraint for mutual information between the inferred variable $z$ and context $c$. This information bottleneck filters down redundant information and compresses the $z$ into a generalisable form. We set the prior $p(z)$ to a unit Gaussian.

$$\phi^* = arg \max_\phi \mathbb{E}_{\mathcal{T} \in p(\mathcal{T})} \left[ \mathbb{E}_{z \sim q_\phi(z|c^{\mathcal{T}})} \left[ \mathcal{G}(\mathcal{T}|z) + \beta D_{KL} \left[ q_\phi(z|c^{\mathcal{T}}) || p(z) \right] \right] \right] \quad (1)$$

Traditional meta-RL methods leverage RNN-based inference systems to produce latent features from a history of recently collected transitions. The problem with this approach is that learning from entire trajectories leads to massive variances which hinder the learning process (Duan et al.; Humplik et al., 2019). In theory recurrence is not strictly required, since any MDP is defined by a distribution of sequentially invariant transitions. In light of this, we exploit task encoders as suggested by Rakelly et al., where $q(z|c)$ is modelled as a product of Gaussian Factors. Each factor $\psi_\phi(z|c_n)$ parameterised by $\phi$, is independently computed: $\psi_\phi(z|c_n) = \mathcal{N}(f_\psi^\mu(c_n), f_\psi^\sigma(c_n))$.

We build on top of probabilistic meta-RL methods derived by Rakelly et al.. Decoupling task inference from task completion allows efficient off-policy objectives to be optimised. Moreover, conditioning a policy with diverse task descriptors aims to generalise behaviours amongst different dynamics and rewards. This is generally not the case when employing MAML-based or RNN-based meta-training kernels to "learn how to learn" new tasks, since policy gradients modify the policy parameters online which, in turn, can result in highly unstable and inefficient adaptation to new tasks, requiring up to hundreds of sample trajectories in unseen environments.

To that end, we employ meta-adapted maximum entropy RL, where a proxy for the policy $\pi(a|s)$ is defined by including the task belief $z$ as part of the observational space $\pi_\theta(a|o, z)$, resulting in task-conditioned critic $\mathcal{L}_{critic}^{\mathcal{T}}$ and actor $\mathcal{L}_{actor}^{\mathcal{T}}$ losses for the the Q-function $Q_\theta(\cdot)$ and the policy $\pi_\theta(\cdot)$ respectively (see A.1 for more details). As VEs are a form of generative processes, $\mathcal{G}(\mathcal{T}|\cdot)$ can be defined with the aim of recovering transition dynamics or reconstructing state spaces (Humplik et al., 2019). Instead, we optimise $q_\phi(z|c)$ in a model-free manner as proposed in Rakelly et al.. Here, parameters $\phi$ can be optimised to maximise expected discounted rewards under a policy $\pi_\theta(\cdot|z)$ through reconstruction of $Q_\theta(\cdot, z)$. We fold $\mathcal{L}_{critic}^{\mathcal{T}}$ as a proxy for the task-dependent goal $\mathcal{G}(\mathcal{T}|z)$:

$$\mathcal{G}(\mathcal{T}|z) \leftarrow \mathcal{L}_{critic}^{\mathcal{T}}(z) \tag{2}$$

### 3.3 Conditioning on Demonstrations

By leveraging meta-learning, we suggest that the agent can learn to exploit the overlapping statistical information from within heterogeneous demonstrations belonging to different tasks, with the aim of inferring task embeddings $z$ which can help improve exploration during test-time adaptation. In order to link demonstrations into the probabilistic meta-RL framework, we formulate an objective based on mutual information $I(z; \tau)$ between the demonstration trajectories and the latent space distribution $p(z)$ (Eq. 3). This objective is similar to that used in Yu et al. with the exception that we adapt this to meta-imitation learning and off-policy RL instead of recovering reward functions.

$$I(z; \tau) = \mathbb{E}_{z \sim p(z), \tau \sim p_\theta(\tau|z)} \big[ \log p(z|\tau) - \log p(z) \big] \tag{3}$$

As direct access to the posterior $p(z|\tau)$ is not available, $I(z; \tau)$ is intractable. However, in order to adjust for this, we leverage a variational approximation to $p(z|\tau)$ using our task belief encoder $q_\phi(z|\tau)$. We assume access to a distribution of expert demonstrations $p_{\pi_E}(\tau|z)$. Additionally, we include further desiderata over the mutual information objective based on distributional matching:

**Learning from demonstrations.** Matching generated trajectories to those sampled from expert trajectories: $\min_\theta \mathbb{E}_{p(z)}[D_{KL}(p_{\pi_E}(\tau|z)||p_\theta(\tau|z))]$. This secondary objective can be considered as trying to match the distribution of trajectories generated by the agent's policy conditioned on $z$, to those from the expert policy (similar to BC). Since they share the same marginal distribution $p(z)$, matching these distributions also encourages matching of the conditionals $p_{\pi_E}(z|\tau)$ and $p(z|\tau)$.

**Linking variational posterior.** $\min_\theta \mathbb{E}_{p_\theta(\tau)}[D_{KL}(p(z|\tau)||q(z|\tau))]$. Encourage $q_\phi(z|\tau)$ to approximate the true posterior $p(z|\tau)$ such that, given a new demonstration, the encoder properly infers the task. This objective acts as a regulator as it reduces over-fitting by penalising variance: optimal distributional mismatch is reached when the encoder produces constant information-less representations of $z$. The distributional matching objective can therefore be defined as in (4).

$$\min_\phi \quad -I(z; \tau) + \mathbb{E}_{p(z)}[D_{KL}(p_{\pi_E}(\tau|z)||p(\tau|z))] + \mathbb{E}_{p(\tau)}[D_{KL}(p(z|\tau)||q_\phi(z|\tau))] \tag{4}$$

After several manipulation steps (see section A.1 for more details), we deconstruct this objective into a differentiable form and define two separate loss components $\mathcal{L}_{bc}^{\mathcal{T}}$ (Eq. 5) and $\mathcal{L}_{info}^{\mathcal{T}}$ (Eq. 6), where $\pi_b$ is the behavioural policy which collected the transitions (truncated importance sampling).

$$\mathcal{L}_{bc}^{\mathcal{T}} = -\mathbb{E}_{\tau \sim p_{\pi_E}^{\mathcal{T}}(\tau), z \sim q_\phi(z|\tau)}[\log \pi_\theta(\tau|z)] \tag{5}$$

$$\mathcal{L}_{info}^{\mathcal{T}} = -\mathbb{E}_{\tau \sim p_{\pi_E}^{\mathcal{T}}, z \sim q_\phi(z|\tau), \tau_b \sim \pi_b(\tau|z)} \left[ \min\left( \frac{\log \pi_\theta(\tau_b|z)}{\log \pi_b(\tau_b|z)}, 1 \right) \log q_\phi(z|\tau_b) \right] \tag{6}$$

As $\mathcal{L}_{info}^{\mathcal{T}}$ conditions the encoder, we aggregate it to the task-dependent goal $\mathcal{G}(\mathcal{T}|z)$ (7).

$$\mathcal{G}(\mathcal{T}|z) \leftarrow \mathcal{L}_{critic}^{\mathcal{T}}(z) + \mathcal{L}_{info}^{\mathcal{T}}(z) \tag{7}$$

### 3.4 Auxiliary Systems

An inherent problem is that the agent generates predictions $Q_\theta(o, a, z)$ based on unsupervised estimates $z \sim q_\phi(z|c^{\mathcal{T}})$. Whilst this is theoretically generalisable to any MDP, it brings large variances and instabilities during training because the agent must simultaneously (i) distinguish one task from another, and (ii) use the task beliefs to solve that task. A plausible solution to mitigate these instabilities is to exploit privileged information (a brief task descriptor such as the position and orientation of a door) during training, allowing the encoder $q_\phi(z|c^{\mathcal{T}})$ to produce task beliefs $z$ which provide succinct descriptors of $\mathcal{T}$. Although this information is not available during testing, it could be made available, for example, if training is performed in a full-state simulation, or in the real-world but with extra environment instrumentation. In theory, a perfect task descriptor is that which provides sufficient information to close the POMDP. We model ground truth task descriptor $\hat{b}$ for $\mathcal{T}$ as a vector $\hat{b} \in \mathbb{R}^v$ where $v$ is the dimension of the task descriptor. During training we wish to condition the encoder to produce latent spaces $z \sim q_\phi(z|c^{\mathcal{T}})$ which, when mapped into the dimensions of $b$, can produce approximations of $\hat{b}$. We use a VAE $d_\lambda(b|z)$ parameterised by $\lambda$ as our auxiliary module, which produces vectors $b_\mu, b_\sigma \in \mathbb{R}^v$ and optimise it via a log-likelihood maximisation objective (8).

$$\mathcal{L}_{aux}^{\mathcal{T}} = -\mathbb{E}_{(\hat{b},c)\sim\mathcal{T}, z\sim q_\phi(z|c)}\left[\log d_\lambda(b = \hat{b}|z)\right] \tag{8}$$

Inference $b \sim d_\lambda(b|z)$ from the trajectories in $c^{\mathcal{T}}$ is decoupled from RL as gradients do not pass through the actor or the critic. This stabilises the encoder by having a fixed supervised target, and allows off-policy training. Leveraging auxiliary objectives for meta-RL was coined by Humplik et al. (2019), where the authors condition task inference solely on $\mathcal{L}_{aux}^{\mathcal{T}}$ to train RNN-based policies. However, in PERIL we use simple descriptors such that these solely condition the encoder to converge towards a suggested direction. Thus, we extend $\mathcal{G}(\mathcal{T}|z)$ with the auxiliary loss to produce the ultimate meta-objective (9). Details of the final computational graph is presented in Figure 8. By keeping the critic loss $\mathcal{L}_{critic}^{\mathcal{T}}$ in $\mathcal{G}(\mathcal{T}|z)$, we give freedom for the encoder to perform unsupervised learning so that it can structure representations of $z$ which represent different dynamics and rewards: if we solely base task inference on $\mathcal{L}_{aux}^{\mathcal{T}}$, a task such as screwing a bolt would output the same task belief as a task of inserting a peg, even though the tasks are fundamentally different.

$$\mathcal{G}(\mathcal{T}|z) \leftarrow \mathcal{L}_{critic}^{\mathcal{T}}(z) + \mathcal{L}_{info}^{\mathcal{T}}(z) + \mathcal{L}_{aux}^{\mathcal{T}}(z) \tag{9}$$

### 3.5 Implementation

Meta-training requires sampling from a set of transitions $(o_t, a_t, r_t, o_{t+1})_{t=0}^T$ corresponding to a distribution of tasks $\mathcal{T} \in p(\mathcal{T})$ and optimising the meta-objective. We use task-specific replay buffers $\mathcal{B}^{\mathcal{T}}$ to store collected transitions from which we can randomly sample during training. Following primal inference objective outlined in section 3.1, task inference is initially conditioned via contextons retrieved from expert demonstrations. Consequently, we store a set of $k$ demonstrations generated by an expert policy on task-conditioned demonstration buffers $\mathcal{D}^{\mathcal{T}}$. This expert policy may consist of a human user or, in our case, a pre-trained SAC agent with access to the full state of the environment. Meta-training algorithms are presented in Algorithms 1 and its process is illustrated in Figure 2. Further details of the meta-training and meta-testing processes in section A.2. We augment $\mathcal{D}^{\mathcal{T}}$ with imperfect trajectories which lead to demo-like (and increased) sum of rewards. This allows PERIL to: (i) be robust to imperfect demonstrations and surpass performance from the expert; (ii) combat the overfitting nature of BC from a narrow set of $k$ demonstrations. The agent must also exploit posterior sampling to update its task belief in order to accomplish continuous adaptation objectives. To this end, we use exploratory data to update the task belief during adaptation and set additional task-dependent encoder buffers $\mathcal{X}^{\mathcal{T}}$ which only stores recently collected contextons. This mitigates distributional shift between off-policy training and on-policy adaptation.

## 4 Experiments

We leverage probabilistic meta-RL methods to find values for $z$ which reconstruct unobserved representations of a new task. To that end, we focus on validating PERIL in environments where the agent must explore in order to understand the dynamics, whilst also utilising sparse rewards to identify the task intent. Furthermore, neural networks, are particularly susceptible to abrupt discontinuities. Thus, it is of particular interest to leverage meta-training in situations as such. We devise

---

**Algorithm 1** PERIL: Meta-training

---

    **Require:** Batch of T training tasks $\{\mathcal{T}_i\}_{i=1,..,T}, \mathcal{T}_i \in p(\mathcal{T})$
1: Initialise $\pi_\theta, Q_\theta, q_\phi, b_\lambda$ and replay $\mathcal{B}_i^{\mathcal{T}}$, encoder $\mathcal{X}_i^{\mathcal{T}}$ and demo buffers $\mathcal{D}_i^{\mathcal{T}}$ for $\mathcal{T} \in \{\mathcal{T}_i\}_{i=1,..,T}$
2: **while** not done **do**
3:     **for** each $\mathcal{T}_i, i \in \{1, ..., T\}$ **do**                       ▷ Collect data
4:         Initialise context from demonstration $c^i = \{\tau_E\}, \tau_E \sim \mathcal{D}_i$ and clear encoder buffer
5:         **for** k = 1,...,$K$ **do**
6:             Sample from context $z \sim q_\phi(z|c^i)$
7:             Roll out policy $\tau \sim \pi_\theta(\tau|z)$. Add to $\mathcal{B}_i$ and $\mathcal{X}_i$. Add to $\mathcal{D}_i$ **if** $\mathbb{E}_R[\tau] > \mathbb{E}_R[\tau_E]$
8:             Update context $c^i \leftarrow \{c^i \cup x^i\}, x^i \in \mathcal{X}_i$
9:     **for** $n = 1, ..., N_t$ **do**                        ▷ Training for $N_t$ steps
10:         **for** each $\mathcal{T}_i, i \in \{1, ..., T\}$ **do**
11:             Sample buffers $x^i \sim \mathcal{X}_i, \tau_E \sim \mathcal{D}_i, b_i \sim \mathcal{B}^i$ to form hybrid context $c^i = \{\tau_E \cup x^i\}$
12:             Sample $z \sim q_\phi(z|c^i)$ and compute losses $\mathcal{L}_{mi}^i, \mathcal{L}_{bc}^i, \mathcal{L}_{actor}^i \, \mathcal{L}_{critic}^i, \mathcal{L}_{D_{KL}}^i, \mathcal{L}_{aux}^i$
13:     Update parameters $\theta_\pi, \theta_Q, \phi, \lambda$ via gradient descent

---

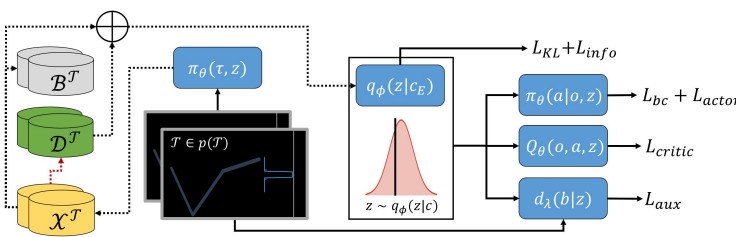

Figure 2: Meta-training framework. Environment contains a set of tasks $\mathcal{T} \sim p(\mathcal{T})$, each with its own ground truth task descriptor $b$. Samples generated by the policy overwrite the encoder buffer $\mathcal{X}^{\mathcal{T}}$ for the current task $\mathcal{T}$. All samples are added to the replay buffer $\mathcal{B}^{\mathcal{T}}$. $\oplus$ concatenates context from the demonstration and $\mathcal{X}^{\mathcal{T}}$ to produce a hybrid context $c_E$. Red dotted line copies collected transitions into the demonstration buffer if these result in higher rewards than the demonstrations.

tasks which involve contact-rich interactions and boundaries, factors which are considered to require a higher level of perception of the environment (Zhu et al., 2020). More details on A.3.



Figure 3: Snapshots of the 5 different task families used. We present two (top and bottom) distinct tasks for each task family: Reach2D, Stick2D, Peg2D, Key2D, Reach3D (left to right). A robotic agent must navigate in order to find the unseen goal. In Stick2D and Peg2D the agent must insert the effector. In Key2D the agent must also rotate the interactive (blue) handle to an arbitrary angle.

The different task families tested in this study are illustrated in Figure 3. Within a task family, one single task is defined by the configuration of the environment, such as the pose of the object or goal. When leveraging privileged information, this pose is used as the task descriptor. For each task, demonstrations are provided from different robot initial poses, and the agent is then required to perform that task, from any new robot initial pose. We compare our proposed methods to other meta-RL and IL baselines. To ensure fairness in all tests, we used the same number of parameters during training. Specifically, MetaIL (Yu et al.), extends PEARL by conditioning exploration on demonstrations. We also use Noisy-BC as a baseline where the agent clones a demonstration with additional noise (20% of $p(o_0|\cdot)$). We also consider the case where privileged information is unavailable, thus compare these baselines to both PERIL and PERIL-A. In PERIL we omit $\mathcal{L}_{aux}$.

### 4.1 PERFORMANCE

We evaluate the performance of each method over 5 task families (Figure 4) using $k = 3$ demonstrations per task. Since meta-learning requires trajectory-based adaptation, we record the averaged trajectory return after 3 policy roll-outs. The results indicate that, in contrast to other baselines, meta-learning without demonstrations through sparse reward feedback is particularly ineffective (PEARL). On the other hand, PERIL-based methods significantly outperform MetaIL and Noisy-BC, especially in tasks from Peg2D or Reach2D/3D where the agent must explore efficiently. It is in these cases where the auxiliary module contributes greatest. On the other hand, in task families such as Stick2D and Key2D, a few contact-rich interactions with the environment can quickly help decipher the task. In this case, goal-oriented auxiliary targets are not as helpful.

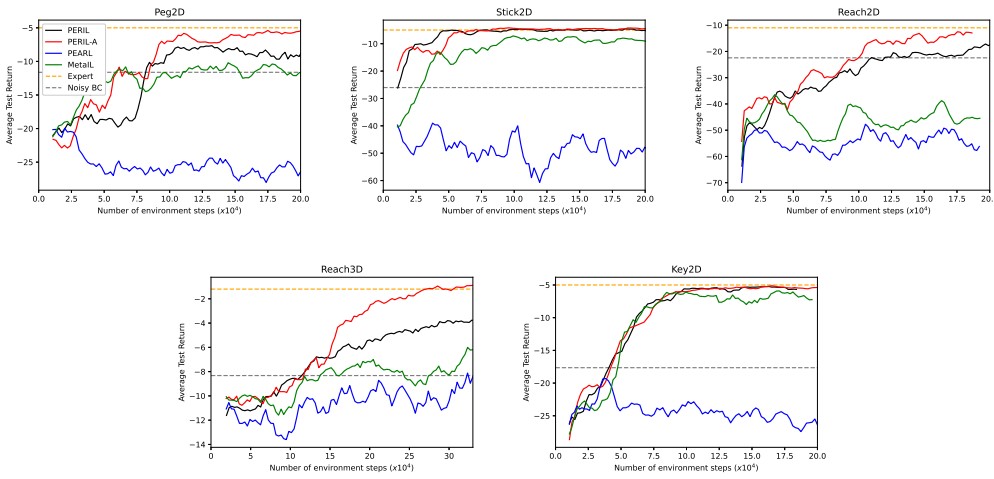

Figure 4: Test-task performance vs. number of collected transitions during meta-training. By exploiting hybrid inference, PERIL consistently achieves better performance with respect to other baselines. Auxiliary systems are particularly useful in complex tasks which require efficient exploration. All results are 3-point averaged.

### 4.2 INCREASING GENERALISATION

Most meta-RL research involves adaptation to tasks within a single task family (Rakelly et al.; Duan et al.; Yu et al.). But in practice, we seek agents capable of interpolating from a diverse set of dynamics. To test the representational power behind PERIL we evaluate training and adaptation performance in multi-task family settings (Figure 5). Here, we train a single agent along each task from from within the 2D task families. TSNE plots of the latent space distribution created during meta-testing reveals PERIL-based methods produce structured task beliefs during hybrid inference of new tasks. Moreover, this verifies PERIL's long-term memory contribution. Last, it showcases PERIL performs structured exploration at the multi-task level, allowing the agent to efficiently switch from one macro policy to another and robustly inherit a diverse set of behaviours. Despite the effectiveness of privileged information in complex exploration tasks in single task family adaptation, auxiliary systems do not generally contribute as much during meta-multi-tasking. We believe that task inference is substantially more convoluted in the multi-task family case as discerning dynamics and intent requires more complex embeddings in the latent space than the position of the goal. We also test generalisation by adapting PERIL in out-of-task distributions (Figure 6), where PERIL interpolates from within meta-trained latent space representations to adapt to unseen dynamics.

Metrics during adaptation of unseen tasks reveal PERIL is substantially more efficient in exploration with respect to other baselines, and is capable of zero-shot learning (Table 1), where the context from the demonstration is sufficient to condition the agent to adapt to an unseen task. Note we record the K-shot as the average kth adaptation trajectory until successful completion of the new task.

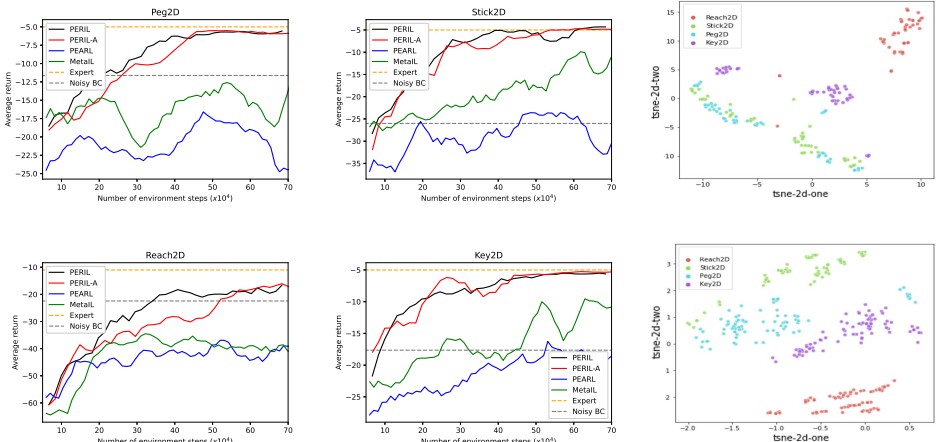

Figure 5: Test-task performance vs. number of collected transitions during meta-training (left and middle). PERIL-based methods are capable of generalising within task families. Latent task distributions $p(z)$ during adaptation (right) demonstrate MetaIL (top) is incapable of clustering tasks within different task families. In contrast, PERIL (bottom) can discriminate them.

Table 1: Mean adaptation rate (K-shot) along all task families. PERIL baselines demonstrate superior adaptation to unseen tasks both in adaptation speed and performance and can endure zero-shot learning.

| Model | K-Min | K-Mean | K-Max |
|---|---|---|---|
| PEARL | 7.2 | $\infty$ | $\infty$ |
| MetaIL | 1.6 | 5.1 | 10.2 |
| PERIL | **0** | 1.0 | **2.7** |
| PERIL-A | **0** | **0.9** | 3.1 |
| Noisy-BC | 1.8 | 8.4 | 12.3 |

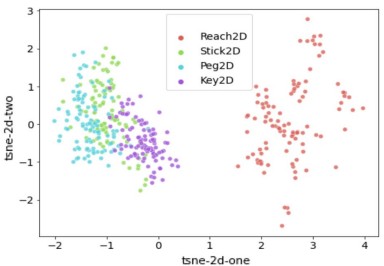

Figure 6: TSNE plot of collected $z$ on adaptation. Stick2D is held out of meta-training and PERIL interpolates dynamics from Key2D and Stick2D.

### 4.3 ALTERATIONS ON K

We study the inherent meta-IL dependency over the number of per-task expert demonstrations $k$. We find that increasing the number of demonstrations per task slightly improves sample efficiency but does not alter the asymptotic capacity of PERIL (Figure 7). Improvement is small since, by augmenting the demonstrations online, PERIL is capable of self-inducing additional demos which improves sample efficiency.

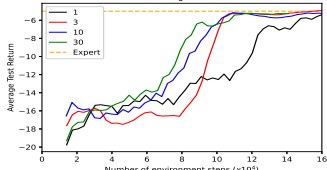

Figure 7: Effect on $k$.

## 5 CONCLUSION

We have introduced PERIL, a new method for meta imitation and meta reinforcement learning which builds a representation of a new task by conditioning on both demonstrations, and further exploration. This provides a framework where new tasks can be defined naturally by a non-expert, without requiring reward shaping or state-space engineering. Experiments across a range of tasks show our method is able to adapt not only to novel instances within a task family, but also to entirely novel task families, whilst doing so with superior data efficiency to a range of baselines. This provides the foundation for further theoretical extensions of this framework, such as learning from imperfect demonstrations, as well as further applications, such as real-world contact-rich robot manipulation.

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

# A APPENDIX

## A.1 MATHEMATICAL EXTENSIONS

**Probabilistic Kernel**

We adopt a MaxEnt RL formulation to define a meta-objective for task $\mathcal{T} \in p(\mathcal{T})$ (10), where $\mathcal{H}(\cdot|o)$ represents the observational entropy, and the temperature $\alpha$ controls the weighting term for the entropy-based reward.

$$\pi^* = arg \max_{\pi} \sum_{\mathcal{T} \in p(\mathcal{T})} \mathbb{E}_{c \sim \mathcal{T}} \left[ \sum_{t \in T} \mathbb{E}_{z \sim p(z|c),(o_t,a_t) \sim \pi^{\mathcal{T}}} \left[ r(o_t, a_t, z) + \alpha \mathcal{H}(\pi(\cdot|o_t, z)) \right] \right] \quad (10)$$

Optimising the objective defined in (10) is possible by using the SAC soft policy iteration approach. The difference to standard SAC is that in our case, the observation is augmented by $z$ resulting in task-conditioned critic (11) and actor (12) losses for the the Q-function $Q_\theta(\cdot)$ and the policy $\pi_\theta(\cdot)$ respectively. $\overline{V}$ corresponds to the frozen target value function (no gradients stored in the forward pass).

Notice we use the term $\mathcal{B}^{\mathcal{T}}$ to represent the distribution of transitions in task $\mathcal{T}$, which is modelled by a replay buffer. Moreover, the over-line operator in $\overline{z}$ denotes that gradients are detached. You may seek further details on the derivation of actor and critic losses in Appendix B of (Haarnoja et al.).

$$\mathcal{L}^{\mathcal{T}}_{critic} = \mathbb{E}_{c \sim \mathcal{T}, z \sim q_\phi(z|c),(o,a,r,o\prime) \sim \mathcal{B}^{\mathcal{T}}} \left[ Q_\theta(o, a, z) - (r + \overline{V}(o\prime, \overline{z})) \right]^2 \quad (11)$$

$$\mathcal{L}^{\mathcal{T}}_{actor} = \mathbb{E}_{c \sim \mathcal{T}, z \sim q_\phi(z|c), o \sim \mathcal{B}^{\mathcal{T}}, a \sim \pi_\theta(a|o,z)} \left[ D_{KL} \left( \pi_\theta(a|o, \overline{z}) \middle\| \frac{\exp(Q_\theta(o, a, \overline{z}))}{\mathcal{Z}_\theta(o)} \right) \right] \quad (12)$$

Where $\mathcal{Z}_\theta(o)$ is the normalising partition function, which is intractable yet has no effect on the computation of the gradients. In the computation of $\mathcal{L}^{\mathcal{T}}_{actor}$, the gradients for $\overline{z}$ are detached, allowing the policy loss to exploit the representation of the MDP passed on by the critic without conditioning the inference network $q_\phi(z|c)$.

Observe in 11 we allow gradients from the inference process $q : C \xrightarrow{\phi} Z$ to pass onto the computation of $Q_\theta(o, a, z)$ such that meta-optimisation of the latent variable $z$ is conditioned on reconstructing the Q-function.

**Conditioning on Demonstrations**

We minimise the conditioned mutual information term by enduring the following deconstruction process.

$$\min_{\phi} \quad -I(z; \tau) + \mathbb{E}_{p(z)}[D_{KL}(p_{\pi_E}(\tau|z)||p(\tau|z))] + \mathbb{E}_{p(\tau)}[D_{KL}(p(z|\tau)||q_\phi(z|\tau))]$$

$$= \min_{\phi} \quad \mathbb{E}_{p(z)}[D_{KL}(p_{\pi_E}(\tau|z)||p(\tau|z))] + \mathbb{E}_{p(z,\tau)} \left[ \log \frac{p(z)}{p(z|\tau)} + \log \frac{p(z|\tau)}{q_\phi(z|\tau)} \right] \quad (13)$$

$$= \min_{\phi} \quad \mathbb{E}_{p(z)}[D_{KL}(p_{\pi_E}(\tau|z)||p(\tau|z))] + \mathbb{E}_{z \sim p(z), \tau \sim p(\tau|z)} \left[ \log q_\phi(z|\tau) \right]$$

In order to optimise the two terms in (13), we define $\mathcal{L}_{bc}$ and $\mathcal{L}_{info}$ as the leftmost and rightmost distribution expectancies respectively. In this definition, the expectancies are not differentiable and requires parametric manipulation. The first step is to adapted this objective into a meta-RL form, by approximating the conditional $p(\tau|z)$ with the policy $p_\theta(\tau|z)$.

We define the first term $\mathcal{L}_{bc}(\theta)$ as the conditional behavioural cloning loss.

$$\mathcal{L}_{bc} = \mathbb{E}_{p(z)}[D_{KL}(p_{\pi_E}(\tau|z)||p_\theta(\tau|z))] \quad (14)$$

By manipulation of the Kullback-Liebler divergence we obtain the following expansion.

$$\min_{\theta} \mathbb{E}_{p(z)}[D_{KL}(p_{\pi_E}(\tau|z)||p_{\theta}(\tau|z)] = \min_{\theta} \mathbb{E}_{z \sim p_{\theta}(z|\tau)}\left[\frac{\log p_{\pi_E}(\tau|z)}{\log p_{\theta}(\tau|z)}\right]$$

$$\min_{\theta} \mathbb{E}_{z \sim p_{\theta}(z|\tau)}\left[\frac{\log p_{\pi_E}(\tau|z)}{\log p_{\theta}(\tau|z)}\right] = \min_{\theta} \mathbb{E}_{z \sim p_{\theta}(z|\tau)}\left[\log p_{\pi_E}(\tau|z)\right] - \min_{\theta} \mathbb{E}_{z \sim p_{\theta}(z|\tau)}\left[\log p_{\theta}(\tau|z)\right]$$

We then substitute policy $\pi_{\theta}$ as the conditional distribution $p_{\theta}(\tau|z)$ and our variatonal inference approximation for the posterior $q_{\phi}(z|\tau)$:

$$\min_{\theta} \mathbb{E}_{z \sim p_{\theta}(z|\tau)}\left[\log p_{\pi_E}(\tau|z)\right] - \min_{\theta} \mathbb{E}_{z \sim p_{\theta}(z|\tau)}\left[\log p_{\theta}(\tau|z)\right]$$

$$= \min_{\theta} \mathcal{H}(p_{\pi_E}(\tau|z)) - \mathbb{E}_{z \sim q_{\phi}(z|\tau)}\left[\log \pi_{\theta}(\tau|z)\right]$$

Because the entropy term is not dependent on the parameters $\theta$, we derive a lower bound objective which is identical to the maximum likelihood estimator for trajectory distributional matching in BC:

$$\mathcal{L}_{bc}^{\mathcal{T}} = -\mathbb{E}_{\tau \sim p_{\pi_E}^{\mathcal{T}}(\tau), z \sim q_{\phi}(z|\tau)}[\log \pi_{\theta}(\tau|z)] \tag{15}$$

Note that in the final representation (15), expert trajectory distributions $p_{\pi_E}^{\mathcal{T}}(\tau)$ are conditioned by task $\mathcal{T}$ as would be the case during meta-training. To obtain a tractable form of $\mathcal{L}_{info}$ we leverage generative processes to switch sampling distributions as we do not have direct access to the prior $p(z)$ to sample $z$. To that end we sample from both expert distributions $\tau \sim p_{\pi_E}^{\mathcal{T}}(\tau)$ and the posterior approximation (encoder) $z \sim q_{\phi}(z|\tau)$ to arrive to (16).

$$\mathcal{L}_{info}^{\mathcal{T}} = -\mathbb{E}_{\tau \sim p_{\pi_E}^{\mathcal{T}}, z \sim q_{\phi}(z|\tau), \tau_{\theta} \sim \pi_{\theta}(\tau|z)}\left[\log q_{\phi}(z|\tau_{\theta})\right] \tag{16}$$

Notice how in $\mathcal{L}_{info}^{\mathcal{T}}$ we generate a set of trajectories $\tau_{\theta}$ from a task belief sampled from expert demonstration trajectories. From this we aim to maximise the likelihood of matching the posteriors on $z$ given $\tau_{\theta}$. In essence, this can be viewed as a two-player game: we want to match the $z$ generated by demonstrations to the $z$ generated by a policy which is also conditioned on the demonstrations.

In this form, $\mathcal{L}_{info}^{\mathcal{T}}$ is incompatible with off-policy RL as $\tau_{\theta}$ is sampled from the current policy $\pi_{\theta}$. We propose truncated importance sampling by sampling from the distribution which collected $\tau_{\theta}$ (17). We use a behavioural policy $\pi_b(\tau|z)$ which is updated every episode as a frozen copy of $\pi_{\theta}$.

$$\mathcal{L}_{info}^{\mathcal{T}} = -\mathbb{E}_{\tau \sim p_{\pi_E}^{\mathcal{T}}, z \sim q_{\phi}(z|\tau), \tau_b \sim \pi_b(\tau|z)}\left[\min\left(\frac{\log \pi_{\theta}(\tau_b|z)}{\log \pi_b(\tau_b|z)}, 1\right) \log q_{\phi}(z|\tau_b)\right] \tag{17}$$

## A.2 Algorithm Details

**Computational Graph**

Figure 8 illustrates the computational graph and training procedure of PERIL.

**Meta Training**

Algorithm 2 is a detailed extension of the meta-training algorithm presented in the main text.

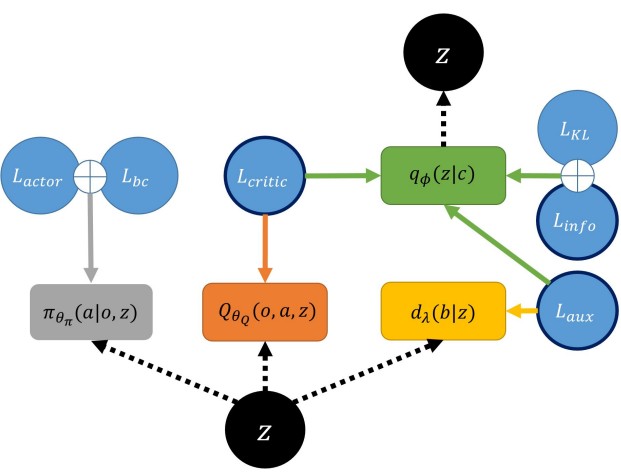

Figure 8: Computational graph of PERIL. Loss gradients (defined by blue circles) propagate to their corresponding models. These include the task encoder $q_\phi$, actor $\pi_{\theta_\pi}$, critic $Q_{\theta_Q}$, and auxiliary module $d_\lambda$. Black circles denote placeholders where latent task beliefs $z$ can be updated (arrow head pointing inwards) or evaluated (arrow head pointing outwards). The operator $\bigoplus$ denotes summation of the loss gradients. The colour of the arrows pointing outwards from the losses indicates the model dependency for that loss. Notice critic and auxiliary losses are shared within two models. Last, the black outlines denote the loss gradients which contribute to the meta-objective $\mathcal{G}(\mathcal{T}|z)$.

---

**Algorithm 2** PERIL: Meta-training

---

    **Require:** Batch of T training tasks $\{\mathcal{T}_i\}_{i=1,..,T}, \mathcal{T}_i \in p(\mathcal{T})$
1: Initialise $\pi_\theta, Q_\theta, q_\phi, b_\lambda$ and $\alpha_\pi, \alpha_Q, \alpha_\phi, \alpha_b$.
2: Initialise replay $\mathcal{B}_i^{\mathcal{T}}$, encoder $\mathcal{X}_i^{\mathcal{T}}$ and demo buffers $\mathcal{D}_i^{\mathcal{T}}$ for $\mathcal{T} \in \{\mathcal{T}_i\}_{i=1,..,T}$
3: **while** not done **do**
4:     **for** each $\mathcal{T}_i, i \in \{1,...,T\}$ **do**
5:         Initialise context by sampling demonstrations $c^i = \{\tau_E\}, \tau_E \sim \mathcal{D}_i$
6:         $\mathcal{X}_i \leftarrow \emptyset$                                       ▷ Clear task encoder to reduce on-policy mismatch
7:         **for** k = 1,...,$K_{prior}$ **do**                       ▷ Demonstration-conditioned trajectories
8:             Sample $z \sim q_\psi(z|c^i)$
9:             Collect trajectories $\tau \sim \pi_\theta(\tau|z)$ and add to $\mathcal{B}_i$ and $\mathcal{X}_i$
10:         **for** k = 1,...,$K_{post}$ **do**     ▷ Exploration-conditioned trajectories via posterior sampling
11:             Sample $z \sim q_\phi(z|c^i)$
12:             $\tau \sim \pi_\theta(\tau|z), \mathcal{B}_i \leftarrow \mathcal{B}_i \cup \tau$
13:             **if** $\mathbb{E}_R[\tau] > \mathbb{E}_R[\tau_E]$ **then**
14:                 $\mathcal{D}_i \leftarrow \{\mathcal{D}_i \cup \tau\}$                               ▷ Demo augmentation
15:             $c^i \leftarrow \{c^i \cup x^i\}, x^i \in \mathcal{X}_i$
16:     **for** $i = 1,...,N_t$ **do**
17:         Sample $R$ random training tasks from $p(\mathcal{T})$
18:         Clone behavioural policy $\pi_b \leftarrow \pi_\theta$                               ▷ To apply TRIS
19:         **for** each $\mathcal{T}_i, i \in R$ **do**
20:             Sample recent trajectories $x^i \sim \mathcal{X}_i, \tau_E \sim \mathcal{D}_i, b_i \sim \mathcal{B}^i$
21:             Form hybrid context $c^i = \{\tau_E \cup x^i\}$
22:             Sample latent task descriptor $z \sim q_\phi(z|c^i)$
23:             Compute losses $\mathcal{L}_{mi}^i, \mathcal{L}_{bc}^i, \mathcal{L}_{actor}^i \mathcal{L}_{critic}^i, \mathcal{L}_{D_{KL}}^i, \mathcal{L}_{aux}^i$
24:         $\phi \leftarrow \phi - \alpha_\phi \nabla_\phi \sum_{i \in R}(\mathcal{L}_{critic}^i + \mathcal{L}_{D_{KL}}^i + \mathcal{L}_{mi}^i + \mathcal{L}_{aux}^i)$
25:         $\theta_Q \leftarrow \theta_\pi - \alpha_\pi \nabla\pi \sum_{i \in R} \mathcal{L}_{critic}^i$
26:         $\theta_\pi \leftarrow \theta_Q - \alpha_Q \nabla_Q \sum_{i \in R}(\mathcal{L}_{actor}^i + \mathcal{L}_{bc}^i)$
27:         $\lambda \leftarrow \lambda - \alpha_\lambda \nabla_\lambda \sum_{i \in R} \mathcal{L}_{aux}^i$

---

**Meta Testing**

Algorithm 3 presentes the process behind meta-testing.

---

**Algorithm 3** Meta-Test adapation

---

**Require:** Networks $\pi_\theta$, $q_\phi$, test task $\mathcal{T} \sim p(\mathcal{T})$, demonstration buffer $\mathcal{D}^{\mathcal{T}}$,
1: Initialise context by sampling $k$ demonstrations $c = \{\tau_E\}_1^k \sim \mathcal{D}^{\mathcal{T}}$
2: **for** r = 1, ..., R **do**
3:     Sample $z \sim q_\phi(z|c)$
4:     Roll out policy $\pi_\theta(a|o,z)$ for $N$ steps and collect trajectory $\tau = \{(o_t, a_t, r_t, o_{t+1})_{t=1}^N\}$
5:     Update context $c \leftarrow \{c \cap \tau\}$

---

### A.3  ENVIRONMENTS

2D task families are set up with Pymunk and the 3D reach task families are set up from MuJoCo's ReachEnv. Each task $\mathcal{T} \in p(\mathcal{T})$ also contains an initial observational distribution $p_0(o)^{\mathcal{T}}$. This distribution defines how the robot is initialised in the environment before collecting data. Since in this project we want to focus on meta-learning the dynamics of contact-rich tasks, we provide an initial distribution $p_0(o)^{\mathcal{T}}$ which allows the agent to start relatively close to the interaction. We summarise $p_0(o)^{\mathcal{T}}$ in Table 2, where $P_0$ and $\theta_0$ represent the initial effector position in Pymunk units (PyU) (or meters if Reach3D) and angle (radians). Note that via inverse kinematics, the other links of the robot are automatically set.

Table 2: Initial observational distribution for all task families, where $E$ denotes the point of interaction and is specific to each task family. In Peg2D and Key2D, $E$ is defined as the clearence hole entry point, where in Stick2D, $E$ is defined by the extrusion's tip.

| Task Family | Distribution |
|---|---|
| Peg2D, Stick2D, Key2D | $100 < |P_0 - E| < 250, |\theta_0| < \pi/4$ |
| Reach2D | $250 < |P_0 - E| < 500, |\theta_0| < \pi$ |
| Reach3D | $0.05 < |P_0 - E| < 0.1$ |

#### A.3.1  SPACES

Each task has an accessible ground truth task descriptor $\hat{b} \in \mathbb{R}^2$ which describes the exact position in space of its goal $G$. In all 2D tasks, the base (origin) of the robotic manipulator is kept constant. The first component of the observational space $\mathcal{O}$ is defined by the normalised relative position from the tip of the end effector ($P \in \mathbb{R}^2$) to the origin $\Omega$ of the robot (18), where $\omega$ is the vector which defines the width and height in PyU of the visualisation domain (1280, 720). The second component of the observational space corresponds to the sine and cosines of the angle of the end effector $\theta$. The action space $\mathcal{A}$ gives complete freedom to the effector and it includes its cartesian velocity $(v_x, v_y)$ (PyU/s) and angular velocity $\omega_\theta$ (rad/s). Through inverse kinematics, the latter are translated back to the other links. Observation and action spaces are summarised in Table 3.

$$o_{rel} = \mathcal{O}(P) = \frac{P - \Omega}{\omega} \tag{18}$$

Table 3: State-action spaces for all task families.

| Feature | Description |
|---|---|
| $\mathcal{O}$ | $\{o_{rel}^x, o_{rel}^y, \cos\theta, \sin\theta\} \in \mathbb{R}^4$ |
| $\mathcal{A}$ | $\{v_x, v_y, \omega_\theta\} \in \mathbb{R}^3$ |

By definition of the observational space, the agent has no information about the position or condition of the goal. Hence, during adaptation the robot must seek to find the goal and identify the task

dynamics using its own interactions as well as the information given by the demonstration. Note that in real world conditions, observation and action spaces could be augmented with kinematic components such as forces and torques at the end effector. However, in this simulated environment, we will show that deciphering the dynamics can be performed by using the aforementioned observational and action spaces.

### A.3.2 REWARD FUNCTIONS

Naive dense reward functions are provided during meta-training of the critic. These reward functions are unshaped and give no information about the dynamics. In essence the reward function $r(\cdot)$ is defined as the absolute distance from the effector tip $P$ to the goal $G$. The only exception is the Key2D task family, where reward functions are augmented with additional incentives to twist the handle to a desired angle $\alpha$. Table 4 summarises the reward functions used in each task family.

Table 4: Dense reward function used for critic meta-training for all task families in MetaSim.

| Task Family | Function |
|---|---|
| Peg2D, Reach2D, Reach3D, Stick2D | $|G - P|$ |
| Key2D | $|G - P| + 0.5|\theta - \alpha|$ |

On the other hand, we only provide binary sparse rewards $r(\cdot) \in [0, 1]$ to the task encoder. These are 0 in all cases except when the effector has reached its goal, where we provide a reward of 1.

### A.4 HYPER-PARAMETERS

Unless otherwise noted, the default hyperparameters for PERIL are as described below.

**Training Loop**

- Action space $|a|$: 3
- Observational space $|o|$: 4
- Latent space $|z|$: 8
- Maximum path length $H$: 60
- Prior collection steps $K_{prior}$: 240
- Posterior collection steps $K_{posterior}$: 240
- RL batch size: 256
- Encoder batch size: 64
- Number of expert demonstrations $K$: 3
- RL sparse rewards: False
- Task belief sparse rewards: True
- Maximum context size: 256
- Number of pre-training steps: 1000
- Number of training steps: 1000
- Number of tasks to average gradients in meta-training $R$: 16
- Number of tasks to collect data from each epoch: 32
- Number of Epochs: 100
- Discount factor $\gamma$: 0.98
- Entropy temperature: 0.1
- Soft target value = 0.005
- Kullback-Liebler divergence weighting $w_{KL}$: 0.1
- Behavioural cloning weighting $w_{bc}$: 0.5

- Mutual information weighting $w_{mi}$: 0.2
- Auxilliary loss weighting $w_{aux}$: 4.0

**Actor**

- Hidden layer size $h$: 256
- Architecture: $[|o, a, z|, h, h, 1]$
- Learning rate $\alpha_\pi$: $3 \times 10^{-4}$
- Output: Bounded action [-1,1] via `tanh`

**Critic**

- Hidden layer size $h$: 256
- Architecture: $[|a, z|, h, h, 1]$
- Learning rate: $\alpha_Q$: $3 \times 10^{-4}$

**Encoder**

- Latent size: $d$
- Input affinity architecture: $[|o, a, r, o|, 256, 256, d]$
- Output affinity architecture: $[d, 64, 64, |z|]$
- Learning rate: $\alpha_q$: $3 \times 10^{-4}$

**Belief Module**

- Task descriptor dimension $|b|$: 2
- Architecture: $[|z|, 32, 16, |b|]$
- Learning rate: $\alpha_b$: $3 \times 10^{-4}$

