# OpenReview forum: "PERIL: Probabilistic Embeddings for hybrid Meta-Reinforcement and Imitation Learning"
_ICLR.cc/2021/Conference — Reject_

### Official Review · AnonReviewer3 · 2020-10-28

**Rating:** 4
**Confidence:** 4

**Review:**

Summary: This work seeks to efficiently learn new tasks by combining meta-RL and imitation learning (IL). Such a combination is a natural thing to try, as both lines of work improve sample complexity of learning a new task: meta-RL by leveraging experience on prior related tasks, and IL by leveraging demonstrations. Demonstrations also form a natural way of specifying a new task to the agent.

This paper extends an existing meta-RL approach (PEARL) to additionally leverage demonstrations. This leads to strong results on a set of 2D problems, as well as a 3D reaching task.

Specifically, PEARL is a Thompson-Sampling approach, consisting of two main components: (i) a learned posterior distribution $q(z \mid c)$, which encodes a distribution over latent $z$’s reflecting the task conditioned on previously observed states, rewards, and actions $c = s_0, a_0, r_0, …$; and (ii) a context-conditioned policy $\pi(a \mid s, z)$, which is used for both exploration to infer the task, by producing more trajectories for the context $c$, and for solving the task, once uncertainty over $z$ is low. Specifically, this paper modifies PEARL in the following ways:
- The context-conditioned policy is trained with an objective similar to behavior cloning to produce trajectories that match the demonstrations, in addition to learning from normal reward signal.
- The posterior $q(z \mid c)$ is trained to produce $z$’s that encode information about the task, given demonstrations for $c$.

Strengths:
- This paper studies an important problem: how can we quickly learn new tasks? For many real-world RL tasks, we want policies that can quickly adapt to new tasks without retraining from scratch. This paper observes that prior approaches have drawbacks: IL on its own can be data-hungry, requiring additional roll-outs or many demonstrations; and meta-RL can be challenging with sparse rewards. Therefore, combining the two is a natural and promising direction to investigate.
- The experimental results are generally quite encouraging. PERIL substantially outperforms the baselines, and can even generalize to a fairly wide distribution of 2D tasks (i.e., the same policy can learn to simultaneously do reaching, peg-placing, and key-rotating tasks, while existing works typically learn a narrower task distribution).

Weaknesses:
- Fairly strong assumption. This paper assumes that an expert distribution $p_{\pi_E}(\tau \mid z)$ over trajectories conditioned on the learned latent $z$ is available. This seems to be a fairly restrictive assumption, since $z$ is learned by PERIL, and therefore, it seems unreasonable for an expert policy to also be able to condition on $z$. Instead, it would be nice if we could relax this to only condition on e.g., the observation $o$.
- Clarity. While the high-level approach is clear, many of the details are confusing and unclear, which makes it challenging to evaluate this approach. I list the main points of confusion below.
    - The problem statement defines the task in terms of $z$, which is confusing, because $z$ should be part of the approach, rather than part of the setting. In particular, it’s unclear what it means for the dynamics model to condition on $z$. It seems like this may be mixing the learned latent $z$ with the true state $s$? More generally, the problem statement (Section 3.1) mixes the approach with the problem setting, which makes it confusing to understand what is a constraint due to the setting, and what’s a design choice for the approach.
    - The principled way to optimize Eq (3) is to maximize variational lower bound (Barber & Agakov, 2003), by substituting the posterior $p(z \mid \tau)$ with an arbitrary function $q(z \mid \tau)$. This appears to be what the paper is doing, but the current phrasing is pretty unclear. In particular, it’s unclear to me how $\mathcal{L}_\text{info}(z)$ is optimized / defined. How is $p(z \mid \tau)$ defined? It’s clear how you can do this in the case where the task descriptor is available, e.g., in Section 3.4, but in general, it’s unclear what the learned $z$ should be. Is this from leveraging the latent space of the expert SAC agent? What is $\pi_b$ in Equation 6?
    - The notation for the task-dependent objective $G(\tau)$ seems unnecessary and serves to distract — in particular, it’s not initially clear why we need this and not just maximizing the expected discounted rewards. I would suggest removing this notation, and just saying at the end of the approach: “overall, we minimize the following loss: $\mathcal{L}_\text{critic}(z) + \mathcal{L}_\text{info}(z) + \mathcal{L}_\text{aux}(z) + \ldots$.
    - There are quite a few undefined loss functions in line 12 of Algorithm 1, in particular $\mathcal{L}_\text{mi}$ and $\mathcal{L}_\text{D_KL}$.
- Related work. This paper generally seems to lack appropriate citations in several key places.
    - In the introduction, several key areas seem to require citations (e.g., citations for meta-IL, posterior sampling with meta-RL should cite PEARL, claims that meta-RL requires shaped rewards / claims that meta-IL cannot adapt afterwards).
    - The following references seem highly relevant to related works section on exploration in meta-RL: [1], [2], [3].
- Experiments.
    - Why is the behavior cloning baseline with noisy demonstrations? It seems like the fair comparison should be BC w/o noise.
    - This paper claims that PERIL is capable of exploring beyond demonstrations, but the tasks that this paper evaluates on don’t seem to require much sophisticated exploration. Substantiating these claims seems to require evaluation on tasks requiring more exploration.

I am initially recommending rejection, due to the aforementioned weaknesses. I believe that the related work and clarity could be improved during the rebuttal period, which would help me raise my score, although I find the strong assumption to be a fairly serious weakness.

Additional comments:
- Ill-formatted citations. Many of the citations are missing the year, e.g., Zhou et al., Ross et al., Mendonca et al., Duan et al., Yu et al.
- Contextons —> contexts?
- Variational Autoencoders are inconsistently abbreviated as VE and VAE. Seems like it should follow the standard of using VAE.

References:
[1] VariBAD: A Very Good Method for Bayes-Adaptive Deep RL via Meta-Learning. Luisa Zintgraf, Kyriacos Shiarlis, Maximilian Igl, Sebastian Schulze, Yarin Gal, Katja Hofmann, Shimon Whiteson. Oct. 2019. ICLR 2020. https://arxiv.org/abs/1910.08348.

[2] Explore then Execute: Adapting without Rewards via Factorized Meta-Reinforcement Learning. Evan Zheran Liu, Aditi Raghunathan, Percy Liang, Chelsea Finn. June 2020. ICML LifelongML Workshop 2020. https://openreview.net/forum?id=La1QuucFt8-.

[3] Environment Probing Interaction Policies. Wenxuan Zhou, Lerrel Pinto, Abhinav Gupta. July 2019. ICLR 2019. https://arxiv.org/abs/1907.11740.

---

> ### Author Response · Authors · 2020-11-23
> **Thank you for your inquiry about the proposed method and the experimental results. Our replies to the questions are listed below.**
>
> Dear reviewer,
>
> We thank you for your valuable feedback and would like to address the following points.
>
> Q: Fairly strong assumption. This paper assumes that an expert distribution $p_{\pi_E}(\tau|z)$ over trajectories conditioned on the learned latent $z$ is available. This seems to be a fairly restrictive assumption, since $z$ is learned by PERIL, and therefore, it seems unreasonable for an expert policy to also be able to condition on $z$. Instead, it would be nice if we could relax this to only condition on e.g., the observation $o$.
> A: Sorry for the misunderstanding. We claim that an expert distribution is available given the true underlying $z$. However, in reality we relax this with an observation, as you suggested. This is indeed something which we should clarify further (since the actor-critic is actually conditioned on an approximation to $z$).
>
> Q: The problem statement defines the task in terms of $z$, which is confusing, because $z$ should be part of the approach, rather than part of the setting. In particular, it’s unclear what it means for the dynamics model to condition on $z$. It seems like this may be mixing the learned latent $z$ with the true state $s$? More generally, the problem statement (Section 3.1) mixes the approach with the problem setting, which makes it confusing to understand what is a constraint due to the setting, and what’s a design choice for the approach.
> A: We need to clarify this further. Indeed the problem setting is based on the true representation $z$. We use inference methods to approximate the distribution in the setting.
>
> Q: The principled way to optimize Eq (3) is to maximize variational lower bound (Barber & Agakov, 2003), by substituting the posterior $p(z∣\tau)$ with an arbitrary function $q(z∣\tau)$. This appears to be what the paper is doing, but the current phrasing is pretty unclear. In particular, it’s unclear to me how $L_{info}(z)$ is optimized / defined. How is $p(z∣\tau)$ defined? It’s clear how you can do this in the case where the task descriptor is available, e.g., in Section 3.4, but in general, it’s unclear what the learned $z$ should be. Is this from leveraging the latent space of the expert SAC agent? What is $π_b$ in Equation 6?
>
> A: We will define the conditional $p(z∣\tau)$. $L_{info}(z)$ is defined in Equation 6 and is derived in the appendices. It is optimised via importance sampling (IS): by sampling latent $z$ from the demonstration context $\tau \sim p_{\pi_E}\mathcal{T}$, we are essentially enforcing a constraint on the similarity between the $z$ represented by a demonstration and the $z$ sampled from the fixed policy $\pi_b$. Notice $\pi_b$ only exists due to IS, and it is defined before Equation 5 (it is the policy which was used to sample trajectories $\tau$ in $\tau_b \sim \pi_b(\tau|z)$. In the computational graph of $L_{info}(z)$,  gradients are passed from $q_\phi(z|\cdot)$. Essentially, this term is a regulariser which controls optimisation of $z$ (critic reconstruction and task descriptor decoding).
>
> C: The notation for the task-dependent objective $\mathcal(G)(\tau)$ seems unnecessary and serves to distract…
> A: We will simplify this as recommended.
>
> C: There are quite a few undefined loss functions in line 12 of Algorithm 1…
> A: Typo which we will correct. $L_{mi}$ is supposed to be $L_{info}$ and $L_{D_{KL}}$ is the Kullback-Liebler divergence from the ELBO optimisation step.
>
> Q: Why is the behavior cloning baseline with noisy demonstrations? It seems like the fair comparison should be BC w/o noise.
> A: We defined Noisy BC as BC with additional “noise” on the action space, which effectively helped it improve exploration at test time. This alternative was not as “deterministic” as normal BC and was therefore a more robust baseline. We will include BC too and/or give further reasons on why we use Noisy BC.
>
> Q: This paper claims that PERIL is capable of exploring beyond demonstrations, but the tasks that this paper evaluates on don’t seem to require much sophisticated exploration. Substantiating these claims seems to require evaluation on tasks requiring more exploration.
> A: We will relax this claim as we intend to raise that, even when the single demonstration is not sufficient to discern the task, the agent is still capable of exploring further to adapt to that unseen task.
>
> We will also like to thank you for your comments which indicate how to improve our related works sections.
>
> We are delighted to hear that, given further clarifications, you would recommend a higher score. Given the reviews from other reviewers, alongside your review, we have acknowledged that we did not have time to make all the necessary changes which would satisfy all of the reviewers. For that matter, we will use all of this feedback to polish our work further and submit again in the following term.
>
> Thank you again and I hope that you find our clarifications and our decision regarding delaying PERIL’s submission appropriate.

---

### Official Review · AnonReviewer4 · 2020-10-31
**A very ambitious work which falls short on the science**

**Rating:** 3
**Confidence:** 5

**Review:**

## Summary of the Work
The work propose a method which allows us to synthesize meta-RL and meta-IL, by pre-training and conditioning a context-based off-policy meta-RL algorithm on imitation data. Strongly inspired by PEARL (Rakelly, et al 2019) and meta-IL (Yu, et al 2020), this method outperforms previous methods by varying margins on a range of newly-introduced 2D and 3D robotics tasks. The work introduces several new design elements and losses to this family of methods, and the experiments do not make clear which ones are responsible for the increased performance. Additionally, it's not clear the included experiments can fully substantiate the long list of claims provided by the authors, not the least of which is that their method performs zero-shot adaptation to new tasks.

## Pros and Cons

### Pros
* Addresses several important problems in adaptive RL
 - generalizing to complex tasks and outside-of-distribution tasks
 - using demonstration data to avoid costly random exploration
 - fine-tuning policies acquired with meta-RL/meta-IL
* Hyperparameters and reward functions are well-documented
* The visualizations are clear and helpful
* Overall the work is well-organized

### Cons
* Makes many claims about the method which are difficult to fully substantiate in such a short paper
* Treatment of prior work is limited to only a few methods from the past few years, and does not acknowledge many prior works in context-based meta-RL/MTRL. Relationship to the very-similar prior works PEARL and meta-IL is unclear.
* Experiments section makes it very difficult to compare presented results to prior work
* Lack of ablations make it unclear which (of many) new design decisions are responsible for the method's performance
* Experiments don't appear to make a credible simulation of demonstration data which might be encountered in the real world
* Claims of applicability to robotics necessitates comparisons with robotics benchmarks (See MetaWorld and/or RLBench).
* Presented method is very complex
* Experiments lack multiple seeds and any statistical significance tests
* Terminology and notation often veers far from previous works and conventions, making it difficult to parse. Sometimes writing is ambiguous or unclear.

## Evaluation
### Quality
3/5
The presented work and experiments are high-quality in their motivation, implementation, and usually their presentation. I think the quality of this work suffers when we consider how well it positions itself with respect to prior work (not well), and the level of detail with which it explores and substantiates each of its claims (of which there are many, making it impossible to address any of them fully). The method section is extensive and recapitulates in detail many concepts from RL, variational inference, IL, etc. It can probably be more sparse to make room for a better treatment of prior work and more experiments/analysis of the work's claims.

### Clarity
2/5
This work suffers from significant clarity issues, mostly around use of language (e.g. zero-shot learning vs few-shot learning), non-standard notation and terminology (e.g. "primal inference," "privileged information," etc.) and the use of equations which are not really necessary to demonstrate a point,

### Originality
3/5
While original in nature (for instance, introducing privileged information, considering fine-tuning and off-distribution tasks, using demonstrations for pre-training, etc.), the work does not make it easy for the reader to divine how its contributions build on significantly-similar previous works which are highly-cited. This does it and the reader a disservice.

### Significance
2/5
If all claims in the work are well-substantiated, it can be a very significant work, but I don't believe they are substantiated. For instance, the introduction mentions low-dimensional observation spaces as a limitation of current meta-RL/IL work, but the experiments don't seem to contain any use of high-dimensional (e.g. image) observation spaces. It's not clear how the claim of zero-shot learning is supported.

### Misc Editorial Comments and Reviewer's Notes

#### Claims
* Addresses three limitations of current meta-RL/IL
 - shaped reward functions
 - constrained low-dimensional action/observation spaces
 - requires hand-defining low-dimensional observation/action spaces
* A hybrid framework which combines the merits of RL and IL
  - tasks defined only using demonstrations
  - unlike other (meta)-IL algorithms, allows for improvement of the policy after adaptation
* Uses only proprioceptive actions from the agent, and implicitly recovers the external environment state
* Allows for learning new tasks without requiring any expert knowledge in the human teacher
* Achieves "exceptional" adaptation rates and is capable of exploiting demonstrations for efficient exploration
* Outperforms other meta-RL and meta-IL baselines
* Is capable of zero-shot learning
* Is capable of multi-family meta-learning and out-of-family meta-learning through clustering in the latent space
* Shows how we can use privileged information (during training) to create an auxiliary loss for training the embedding function, allowing us to recover the "true underlying state"

#### Mechanisms
* Represents a task by a latent vector, which is the belief state of the task given a demonstration
* Meta-training encourages high mutual information between the demonstration data and latent space
* After adaptation, the agent can explore and update the latent space

#### 1. Introduction
* "hand-crafted, shaped reward functions..." nothing in the meta-RL formulation requires shaped reward functions, as opposed to a sparse ones which are easier to craft. Granted, on-policy meta-RL algorithms are challenges by sparse reward settings in a similar fashion that on-policy RL algorithms are challenged by sparse reward settings, but this is a property of RL in general and not just meta-RL. [2] is a meta-RL method which can cope with sparse rewards, and is extensively-cited in this work.

* "defining a low-dimensional.." this is not a limitation per se of meta-RL or meta-IL -- there's nothing in their formulation which necessitates meta-RL operating on low-dimensional state as opposed to images, though it is certainly a design challenge. See [1]

* "different task families" Perhaps I have misunderstood the authors, but this seem orthogonal to the purpose of the work and of meta-RL. It is not immediately clear what the authors mean by "task families." While there is certainly work on cross-domain transfer in IL and RL, adapting policies to different action and/or observation spaces is not a typical goal of meta-IL and meta-RL algorithms, so it seems strange to level this critique.

[see "claims" above]

#### 2. Related Work
* Meta-learning and meta-RL far predates Wang and Duan. Please see [3,4, etc.]
* Modern work on context-based meta-RL and adaptive RL predates PEARL. Please see [5, 6, 7] which all perform variational inference on trajectories to generate a latent context, which can then be used for adaptation

#### 3. Method
* The proposed approach seems hardly different than PEARL[2] with the following changes. The reviewer may have missed something, but given close relationship between these methods, please make crystal clear for readers the differences between this method and the substantially-similar PEARL method.
  - Pre-training uses demonstration data rather than RL episodes
  - This work studies what happens if you continue training after adaptation
  - Introduces an auxiliary loss which allows conditioning on privileged information
* 3.2: "Traditional meta-RL methods leverage RNN-based" - This is hardly true in any universal sense. Previous meta-RL method have used RNNs [8], variational inference [2], autoregressive models (attention)[9], hierarchy [10], exploration policies [11], etc. to implicitly model the latent task space.
* 3.3: The included equations don't seem to add much to the paper's story and seem to recapitulate well-known results from RL, variational inference literature, or cited work.
 * 3.5: The use of a SAC expert trained on full-state versions of the environment is not a faithful simulation of expert data, which will be significantly noisier and lower-entropy than a SAC expert, and also is unlikely to be optimal according to any RL loss. I think that this reduces the IL aspects work to a form of offline RL where the offline data source is a SAC policy, and the results demonstrate that the method can reconstruct the privileged information which was available to the SAC agent. The authors note that they augment the demonstration data with "imperfect demonstrations", but are silent about how this is achieved (and it must be done with great care).

#### 4. Experiments
* 4.1: Though it is ambiguous from the text, these experiments seem to either present 1 experiment per method, or the average of 3 experiments per method. This is unfortunately not enough data to make a statistically meaningful comparison of the performance, especially considering the small performance differences involved. Please see [12] for a handy guide on how to compare performance. In short, you will likely need 10 seeds for each experiment and should conduct a statistical test to ensure your differences are real. Please include a 95% confidence interval in your plots for the benefit of readers.
* 4.1: These experiments are meaningful and helpful, but it's also important to readers that they can verify you have reasonable implementations of the comparison methods. This necessitates providing some results for some of the environments used in Yu, et al and/or Rakelly, et al. How is the reader to know your implementation or hyperparameters are fairly representing the comparison methods?
* 4.2: I think the reader would benefit from seeing t-SNE plots from the comparison methods as well. The presented plots look very similar to t-SNE plots generated by plotting samples from a PEARL posterior.
* 4.2: It is very unclear what the authors mean by "zero-shot" learning. By my estimation, this method always requires some samples of the target environment to attain the presented performance, making it squarely a few-shot domain.
* This work introduces many new design elements on top of PEARL and Yu et al, and it's unclear which of them are responsible for the observed performance. Please include ablations which compare your method's performance without each new design element, to demonstrate the impact of each.

[1] https://arxiv.org/pdf/2006.07262.pdf
[2] http://proceedings.mlr.press/v97/rakelly19a/rakelly19a.pdf
[3] http://citeseerx.ist.psu.edu/viewdoc/summary?doi=10.1.1.34.1796
[4] https://www.sciencedirect.com/science/article/abs/pii/S0893608002002289
[5] https://openreview.net/pdf/29c35690ff52463c84d9456ab511e4c944ddbea4.pdf
[6] https://arxiv.org/pdf/1809.10253
[7] https://arxiv.org/abs/1806.02813
[8] https://arxiv.org/abs/1611.02779
[9] https://arxiv.org/abs/1707.03141
[10] https://arxiv.org/abs/1710.09767
[11] https://arxiv.org/abs/1802.07245
[12] https://arxiv.org/abs/1904.06979

---

> ### Author Response · Authors · 2020-11-23
> **Thank you for your inquiry about the proposed method and the experimental results. Our replies to the questions are listed below.**
>
> Dear reviewer,
>
> We would like to thank you for your detailed feedback. We see this as an opportunity to clarify our results and our methodology,  as well as chance to present further studies which may concretely point out the contributions of our approach. We are happy to hear that, if well substantiated, you believe we could be delivering some very significant work. In light of this we are aware of the changes required to polish our paper and we have thus decided to postpone our submission and use your feedback, along with that from the other reviewers, accordingly.
>
> We would like to thank you for your comments, and will pay particular attention to:
> - Improving our review on meta-RL and meta-IL methods
> - Clarifying the differences with PEARL
> - Clarifying how we include imperfect demonstrations into the demonstration buffers (which theoretically make IL more robust to high-entropy demonstrations).
> - Include confidence intervals for the experiments
>
> We would like to clarify the following issues.
>
> Q: "different task families" Perhaps I have misunderstood the authors, but this seem orthogonal to the purpose of the work and of meta-RL. It is not immediately clear what the authors mean by "task families." While there is certainly work on cross-domain transfer in IL and RL, adapting policies to different action and/or observation spaces is not a typical goal of meta-IL and meta-RL algorithms, so it seems strange to level this critique.
> A: We seek for agents which can interpolate from within tasks from different task families. For instance, if a robot can insert a peg and sharpen a pencil, why can't it adapt to screwing a bolt? In our experiments we evaluate this only within agents which have the same observational and action spaces. For instance, 2D agents do not cross-platform (multi-task meta-learning) with 3D agents.
>
> C: "3.3: The included equations don't seem to add much to the paper's story and seem to recapitulate well-known results from RL, variational inference literature, or cited work."
> A: Some reviewers have noted that we cannot assume VAE is a commonailty. We believed that including terms for how the meta-encoder parametrised meta-learning would be important for the reader.
>
> Q: "4.1: These experiments are meaningful and helpful, but it's also important to readers that they can verify you have reasonable implementations of the comparison methods. This necessitates providing some results for some of the environments used in Yu, et al and/or Rakelly, et al. How is the reader to know your implementation or hyperparameters are fairly representing the comparison methods?"
> A: We use the same hyperparameters and approximately the same parameters for all training algorithms (when possbile): "To ensure fairness, we use the same number of parameters in all tests". We will clarify this further (perhaps in the appendix).
>
> Q: "4.2: It is very unclear what the authors mean by "zero-shot" learning. By my estimation, this method always requires some samples of the target environment to attain the presented performance, making it squarely a few-shot domain."
> A: It is unfortunate that we did not clarify this enough. We believe this is our strongest result (and contribution): by giving PERIL a single demonstration of an arbitrary task from within i.e. the 4 2D task families, it can perform that task successfully without any adaptation stages (In contrast to many other methods).

---

### Official Review · AnonReviewer2 · 2020-10-31
**Writing is not clear and needs some work, Connection to prior work needs extra attention, An additional baseline is needed**

**Rating:** 4
**Confidence:** 4

**Review:**

## Summary of work
This work proposes PERIL, a method for combined Meta Imitation Learning and Meta Reinforcement Learning using context-based meta-learning. Given a set of demonstrations, a latent variable representing the desired task is inferred, and trajectories are generated conditioned on the inferred latent variable.  The data from the expert demonstrations and trajectories are used for meta-learning updates.

## Review
Key comments below are included in bold text.

* Related Work
  * I don't agree that meta-learning was conceptualized as an RNN-based task. There are many variants of meta-learning, RNN-based, MAML (optimization based), Encoder-based (Neural Processes), etc.
  * It seems that "Scalable Meta-IRL Through Context-Conditional Policies" (Ghasemipour et al. 2019) and maybe "Robust Imitation of Diverse Behaviors" (Wang et al., 2017), are closely related context conditional meta learning methods (similar to Yu et al. which you cited). They may merit citation.
  * Bottom of page 2: Why do they have the traditional caveats? They are using rewards, so they should be able to do better.
* Section 3
  * Section 3.1: A number of works which also combine Meta-IL & Meta-RL use the setup you use as well (Primal Inference + Exploratory Adaptation). I think you should acknowledge this and provide citations (such as Mendonca et al., Zhou et al., Gupta et al. which you cited in other sections).
  * Section 3.2: What is a Variational Encoder? This is not a standard term.
  * Section 3.2: Please clarify how the KL relates to the mutual information. While this may be a simple connection, I think it should be explained better.
  * __Section 3.2: Where is equation 1 coming from? It looks like the Evidence Lower Bound but $\mathcal{G}(\mathcal{T}|z)$ is not specified and could be a non-likelihood objective function. Please clarify.__
  * __Section 3.2: You mention that you are using an encoder in the same manner as Rakelly et al., but you do not actually describe the method in your manuscript and how you perform context aggregation. Please include a complete description in the main manuscript.__
  * __Section 3.2: First paragraph of page 4 requires significantly better clarification. Specifically the sentences "This is generally not ... in unseen environments." It is not clear why you are saying your context-based approach is better.__
* __Section 3.3:__
  * __While you do acknowledge Yu et al. in the beginning of this section, you understate the extent to which this is similar to their mutual information objective. Equations 3&4, "Learning from demonstrations", and "Linking variational posterior" are effectively identical to Yu et al.'s with different variable names. You should clarify and attribute credit to their work in a more clear manner.__
  * __I do acknowledge that you have tried to address the intractability of $\mathcal{L}\_{info}$ in a different manner than Yu et al., but the main manuscript would benefit from a brief explanation for why you took this route (even though you have derivations in the appendix)__
  * __You need to explain why $\mathcal{L}\_{BC}$ is not included in your objective. From looking around in your appendix, I think this is because you are estimating the $p(z)$ marginal with the expectation distribution as in equation 5, so you don't take the gradient with respect to $q\_\phi$. This still needs explanation in my opinion.__
  * __Section 3.4: First sentence, what do you mean by unsupervised? You are using the critic to train the model the generates $z$. Please elaborate.__
  * __Section 3.4: Sentence before equation 9, doesn't this depend on what information is included in the vector $b$?__
  * __General Question: What is the reinforcement learning algorithm you are using?__
  * __Section 3.5: I don't understand what you mean by $\mathcal{D}^{\mathcal{T}}$?__
* Section 4
  * Please explain what the auxiliary informations are for each task.
  * __Baselines__
    * __As far as I can tell, you have not compared to any prior method that combines Meta-IL and Meta-RL (such as Mendonca et al. or Zhou et al. which you cite in your work). I think this is a big flaw of your experimental section. Including this would demonstrate whether your latent-based meta-learning approach is better or not from prior works with used other types of meta-learning, which should be your main value proposition. I think latent-based might work better than the MAML-based as in some prior work, but you have not provided any experiments to support this.__
    * __Why are you comparing to noisy BC and not normal BC (also from your description of noisy BC is not clear what the method is)? Also please clarify what BC loss you are using (e.g. mean-squared error loss, maximum likelihood with learned variances, or whatever you used).__
    * __I think it would be valuable to also include one setting where only the auxiliary loss + critic loss (no mutual info) is used to see the the value of the additional mutual information losses.__
  * Section 4.1: How many train/test tasks in each task family.
  * __Section 4.1: You mention sparse rewards here, but in the appendix there is information about "dense rewards during meta-training" for the critic. I don't quite understand what is happening here. Are you using sparse or dense rewards? And is every baseline method that uses rewards use the same type of reward? Please clarify.__
  * __Section 4.2: Do you mean that you are training a single agent on all tasks simultaneously? How are you doing this when in the observation/action spaces are different? Do all these have the same observation/action space?__
  * __Section 4.2: Last sentence of paragraph 1 needs elaboration. I don't think you have provided any explanation for these experiments, for example how you are setting up experiments for adapting to unseen dynamics.__
  * __Section 4.2: Table 1, how is 0 possible? Also, you haven't provided explanation for how these values are computed.__

---

> ### Author Response · Authors · 2020-11-23
> **Thank you for your inquiry about the proposed method and the experimental results. Our replies to the questions are listed below.**
>
> We have acknowledged that we require further experimental validation via additional baselines, in addition to clarifying and identifying the contributions of each component in PERIL. For that matter, we have decided to use your feedback amongst that from the other reviewers and postpone our submission.
> In no lesser extent, we are grateful for you valuable feedback which we will most definitely use to shape our new submission. We thank you for your comments which you have raised and hope that you find our answers (see below) and our decision appropriate.
> We will look into:
>
> - Changing the historical introduction to Meta-Learning, introducing further studies regarding meta-IL (we accidentally used the wrong citation for “bottom of page 2”), and further accrediting other authors with contributions to the field
> -Clarifying the problem formulation further (VAE, ELBO)
> -Clarifying why $L_{BC}$ is not included in our objective (reasons are as pointed out)
> -Adding other baselines which clarify contributions of each component of PERIL
>
> We also wanted to answer and/or ask you some questions about some of your concerns which we did not quite understand.
>
> Q: “You mention that you are using an encoder in the same manner as Rakelly et al., but you do not actually describe the method in your manuscript and how you perform context aggregation. Please include a complete description in the main manuscript.”
> A: By context aggregation do mean the Gaussian factor approach they used in Rakelly et al., or the hybrid context using demonstrations and collected transitions?
>
> Q: “Section 3.4: What do you mean by unsupervised? You are using the critic to train the model the generates $z$. Please elaborate”
> A: We aim to point out that $z$ is based on the estimates of the critic network which initially have high variances and are unstable, which is why a supervised objective brings stability. We will clarify this.
>
> Q: Section 3.4: Sentence before equation 9, doesn't this depend on what information is included in the vector $b$?
> A: Yes, it depends on what we include in $b$ but in order to make our model more adaptable, we suggest that this should only be used as an auxiliary loss function. For example, in Stick2D and Peg2D, we use the same value for $b$ but the tasks are different.
>
> Q: General Question: What is the reinforcement learning algorithm you are using?
> A: SAC. We mention this in the appendices but we could benefit from mentioning this in the main manuscript.
>
> Q: I don't understand what you mean by $D^\mathcal{T}$?
> A: Defined in section 3.5, these are task-related demonstration buffers, where we store the $k$ demonstrations of each task.
>
> Q: Section 4.1: You mention sparse rewards here, but in the appendix there is information about "dense rewards during meta-training" for the critic. I don't quite understand what is happening here. Are you using sparse or dense rewards? And is every baseline method that uses rewards use the same type of reward? Please clarify.
> A: We are training the SAC with dense rewards but the context collected (which is used by our encoder to produce $z$) receives sparse rewards so that, at test time, it experiences less distributional shift from one MDP to another. All the baselines use the same type of setting as a fair test.
>
> Q: Training and testing tasks?
> A: Section 4.1 : We have 40 training and 10 testing tasks for all 2D task families and 90 training and 10 testing tasks for the 3D MuJoCo task family. We will include this info in the manuscript.
>
> Q: Why are you comparing to noisy BC and not normal BC (also from your description of noisy BC is not clear what the method is)? Also please clarify what BC loss you are using (e.g. mean-squared error loss, maximum likelihood with learned variances, or whatever you used).
> A: We include Noisy BC (BC + random noise on the action) because it is better at searching the space than just BC. We can also add normal BC as a comparison. We will also clarify details for our BC loss further.
>
> Q: Section 4.2: Last sentence of paragraph 1 needs elaboration. I don't think you have provided any explanation for these experiments, for example how you are setting up experiments for adapting to unseen dynamics.
> A: All of the agents have the same observation and action spaces. They have to decipher (1) the task family, and (2) the task. Regarding the last sentence of paragraph 1, we are doing so by training on task families Peg2D, Reach2D, Key2D and testing on Stick2D. These have different dynamics.
>
> Q: Section 4.2 Table 1, how is 0 possible?
> A: Table 1. 0 is possible because it takes 0 adaptation rollouts (which we will now call $K_\tau$ as suggested by Reviewer 5) to succeed at performing this task. These values are computed by finding at which rollout $K_\tau$ the agent successfully completes the task, given that the next 2 attempts will also be successful. We are aware that this was not clarified given the 8-page constraints but we will attempt to clarify this.

---

### Official Review · AnonReviewer5 · 2020-11-06

**Rating:** 4
**Confidence:** 3

**Review:**

Summary:

This paper introduces PERIL, a meta RL method that combines demonstration trajectories and trajectories collected by the policy, in order to adapt to a new task. To this end, the authors combine ideas from metaRL (specifically from PEARL (Rakelly et al. 2019) and Humplik et al (2019)) where a set encoder is used to encode trajectories to a latent vector describing the task, with imitation learning techniques by (a) training this encoder also with demonstrations (b) initialising the latent vector at test time by feeding demonstrations through the encoder, and (c) having additional losses inspired by metaIL techniques. The motivation is that using demonstrations allows us to learn tasks that are difficult otherwise, for example because the rewards (at test time) are sparse.

Overall impression:

I like the idea of using demonstrations for metaRL when tasks are sparse. Many metaRL methods do not work well in sparse reward tasks, and using expert demonstrations is a nice way of guiding the agent towards behaviour that can solve the task. Empirically, the proposed method PERIL outperforms the baselines PEARL and MetaIL, so that is promising. The authors provide analysis of the latent space which nicely illustrates what the method has learned. However, PERIL is quite complex since it consists of many different parts and loss terms (six if I counted correctly), and it needs demonstrations + interactions + (sparse) reward signals at test time. I found it hard to keep track of everything and make sense of how these parts fit together. From both the text and the empirical results, it is not clear to me why all the parts are necessary / what they do, and I am left wondering if a simpler approach would work as well. The notation and mathematical formulation in the paper is not polished enough (there are inconsistencies, variable name clashes, some parts of the objective function not properly introduced and explained) which added to my confusion. Therefore, even though the idea seems promising, I think the paper is not quite ready for publication.

Questions:
- In the introduction you say that MetaIL methods have the drawback that "after adaptation, they cannot continue to improve the policy in the way that RL methods can". You say that you method PERIL "allows for continual policy improvement through further exploration of the task". I have a few questions about this.
 - Since only the latent embedding is updated, doesn't PERIL also suffer from the fact that the policy cannot be improved in the way that RL methods can (but instead, all adaptation is that within the limits of task inference)?
 - Why is additional exploration at test time even necessary, if we have expert rollouts and the policy itself isn't actually updated (the only thing that's adapted at test time is the latent embedding)? If all the demos + trajectories are used for is task inference, then shouldn't the expert demos always be sufficient?
 - You motivate your approach by saying that at test time, it is useful if the expert does not have to provide a shaped reward. However, you do make use of a shaped reward during training - this is a limitation that should be discussed in the paper. In addition, you still need (sparse) rewards at test time. Are those really necessary, given that you have a demonstration of the task? Did you test PERIL without those sparse reward inputs to the encoder?
- Table 1, how was the agent trained? Was it with number of adaptation trajectories k>0? If so, what if you would train the agent with k=0? On the other hand, can you get good zero-shot adaptation performance by just increasing the number of demonstrations?
- You say $d_\lambda$ is a VAE, but if I understand your setup correctly then $d_\lambda$ is only the decoder of a VAE right? And Eq 8 is the reconstruction loss? Which also means it's not technically a VAE, because in encodes and decodes different things (encodes trajectories, decodes task descriptors - Humplik et al. 2019 describe this as an information bottleneck). Shouldn't there also be a KL term somewhere here?
- Where is $L_{bc}$ used? It's not part of Eq (7), but I also can't find it anywhere else except in Algo 1 and Fig 2. And what about $L_{mi}$, it's only in Algo 1 but nowhere else? Fig 2 has $L_{KL}$, where is that from? It would really help my understanding of piecing everything together if there was one single equation somewhere, that includes all loss terms. For each loss term, I as the reader want to clearly understand where it comes from, and why it is necessary (see suggestions for additional baselines/ablations below).

Suggestions / Feedback:
- The problem formulation and the proposed solution don't match. In your problem setting you say you're in a general POMDP where the true state may be partially observed, but in your algorithm you rely on the fact that it's a POMDP only w.r.t. the task (i.e., reward and transition function) and *not* w.r.t. the environment state. That's an important difference! To explain that in more detail: in the introduction and problem setting you say $z$ models the true underlying state $s$ which can change at any moment: your transition function is $p(o',z'|o,a,z)$ where $o$ are observations. However, the entire formulation in your algorithm relies on the fact that $z$ does _not_ change over time, but instead describes a fixed task. That's also what PEARL does, which is what your formulation is based on. I think there's two ways to resolve this: (A) Either change the problem setting such that $z$ is fixed throughout time and define the transition function as $p(s'|s,a,z)$ where the environment state $s$ is now fully observable, or (B) change the algorithm to actually model a belief over a latent $z$ that can change over time. Option (A) is probably an easier fix, but then you might also have to change some of the environments (if I understand correctly, in Key2D the state of the handle is unobserved and can change the unobserved environment state).
- Section 3.1, I would add explicitly what the objective of the policy is (both in writing and in a mathematical expression). You aim to maximise the return of a policy that conditions on $K_d$ demonstrations, and which has interacted with the environment for $K_r$ rollouts (changing the notation here to make the distinction clear). From there it is easy for the reader to see what happens if you set $K_d=0$ (you get something more similar to PEARL), and what happens when you set $K_r=0$ (which is the zero-shot case). It's good to contrast this for the reader, and explain / show empirically why and when $K_d>0$ and $K_r>0$ is necessary. (See my comment on baselines below.)
- To understand PERIL better, I would suggest to add a few baselines.
 - PEARL with a pre-initialised buffer that contains the demonstrations. The encoder and policy will be trained as normal, but there's additional data coming from the buffer that contains expert trajectories. Since PEARL uses an off-policy algorithm it is possible to train the policy with this data. I think this is an important comparison, because it's a very simple way to incorporate demonstrations into PEARL and it would be good to understand if/when/why this works/doesn't work.
 - In addition to the above, use the demonstrations at test time to initialise the context in PEARL. This is very close to the setting in PERIL, except that some parts of the objective function are missing ($L_{info}$, $L_{aux}$, $L_{bc}$, $L_{mi}$ - I think).  This would give insight into whether those additional losses are truly necessary (currently you only have ablations on $L_{aux}$).
 - Zero-Shot PERIL. There is some analysis of this in Table 1, but I think it would still be helpful to add this baseline. Does it work well for within-task-distribution adaptation (Fig 4) and not so well for settings that require more generalisation (Fig 5)? What if we just throw in more demonstrations, is that sufficient to do zero-shot adaptation or do we really need the policy rollouts? I think this is a central question that should be very clearly answered in the paper. Table 1 is a good start but this analysis can be expanded.
 - Humplik et al. (2019), but with additional demonstration data to train the encoder/decoder. Again, this is the simplest way to incorporate the demonstration data into this method without explicitly making use of it at test time. This comparison would tell us something about why the demonstrations are necessary - are they necessary during training but not at test time, or the other way around, or are they necessary both during training and testing?
 - Not sure I got everything, there's still $L_{bc}$ and $L_{mi}$ which I'm not entirely sure where they come from and if they are necessary. But basically, I think it's really important to analyse which parts are necessary - and make the method as simple as possible if you find some parts are not necessary.

Smaller comments (didn't influence my score):
- There's a clash between using the variable k/K for the demonstrations (e.g., Sec 3.1 "primal inference", Sec 4.1 first sentence, Fig 7), and for the number of policy rollouts (e.g., algorithm 1 line 5, Table 1). This is confusing, so I strongly suggest using two distinct variable names (something like $k_d$ and $k_r$ also works).
- Similarly it would help if you use two separate notations for the trajectories $\tau$ that come from the demonstrations, and the ones that come from the policy. Throughout Section 3 I don't always understand which one of the two you are talking about.
- Your references need fixing. Some of them are without a year, and some say technical report even though they were published at a conference (e.g. Finn et al., Rakelly et al.). Your "Wang" reference for RL2 also seems wrong (first sentence in related work)? It should be Jane Wang et al. "Learning to reinforcement learn". The way I always get my bibtex entries is via scholar.google.com: search for the paper there; click on the "cite" button under it and then "BibTeX" (Double check whether the paper was published somewhere though, google scholar often only puts the arXiv link then actually the paper was published somewhere. Sometimes the authors also put the correct bibtex comment on their homepage/github with the code).
- Fig 1 left, there's a typo: "learn to lean" -> "learn to learn"
- For your experiments, I would call PERIL-A only "PERIL" (since this is your full method, including all losses), and then call the *ablations* different, so for example PERIL-noAux when you remove the auxiliary loss.
- All figures should have some form of indication of the error/std/confidence interval (using shaded regions around the mean for example).
- Sec 4.2, explain what the multi-task family setting is and why it is challenging.

---

> ### Author Response · Authors · 2020-11-23
> **We thank reviewer 5 for their constructive feedback and suggestions.**
>
> Dear reviewer,
>
> We deeply appreciate the effort you have put into reviewing our paper. We take your feedback as an opportunity to further develop evidence which supports our proposed method. Given the time constraints during the rebuttal period, we have decided to use your feedback to present a more thorough submission of PERIL in the future. In particular, we thank you for:
>
> - Your comments on how to problem formulation: we will stick to option A and will have Key2D as an example of a system which requires demonstrations to rapidly condition exploration.
> - Your views on how to clarify the problem by adding $K_d$ demonstration and $K_\tau$ rollout terms.
> - Your indications on which baselines we could add to further understand the contributions of the different components in PERIL.
>
> We also thank you for other comments which contribute to our understanding to which areas (especially in our methods section) require further clarification.

---

### Decision · Program_Chairs · 2021-01-07
**Final Decision**

**Decision:**

Reject

**Comment:**

The paper proposes a method that combines imitation learning and meta-learning, which aims to be able to explore beyond the provided demonstrations.

While the paper addresses an important topic, and the authors are commended on a productive conversations, there is a consensus among the reviews that the work is not yet ready for publication. The future manuscript should address: reexamine the assumptions and improve presentation.